

# Age, growth, reproduction and mortality of *Xenocypris argentea* (Günther,1868) in the lower reaches of the Tangwang River, China

Peilun Li[1,2], Jiacheng Liu[1,3], Wanqiao Lu[1,2], Shuyang Sun[1,4] and Jilong Wang[1,2]

[1] Heilongjiang River Fisheries Research Institute, Chinese Academy of Fishery Sciences, Harbin, China
[2] Scientific Observing and Experimental Station of Fishery Resources and Environment in Heilongjiang River Basin, Ministry of Agriculture and Rural Affairs, Harbin, China
[3] College of Animal Science and Technology, Northeast Agricultural University, Harbin, China
[4] College of Fisheries and Life Science, Dalian Ocean University, Dalian, China

Corresponding author
Jilong Wang, wangjilong@hrfri.ac.cn, wjl0321225@163.com

## ABSTRACT

To investigate various population biological parameters of *Xenocypris argentea* in the lower reaches of the Tangwang River (China), a comprehensive study was conducted for the first time. A total of 1,003 samples were collected from April to November 2022. The collected samples revealed that female *X. argentea* had total lengths ranging from 12.4 cm to 25.7 cm (weighing 15.86 g to 159.55 g), and male *X. argentea* had total lengths ranging from 10.8 cm to 23.9 cm (weighing 9.27 g to 121.06 g). The age of the samples was determined using otolith analysis, indicating that the ages ranged from 1 to 5 years old in both females and males. The length-weight relationships were further analyzed, uncovering the allometric growth index ($b$) was 3.1296 for females, indicating a positive allometric growth pattern. Differently, males exhibited a $b$ value of 3.0274, suggesting an isometric growth pattern. Furthermore, the von Bertalanffy growth formula provided insights into the growth characteristics of *X. argentea*, revealing an asymptotic total length ($L_\infty$) of 36.096 cm and a growth coefficient ($K$) of 0.121. The analysis of the gonadal somatic index ($GSI$) and ovarian development period indicated that the spawning period occurred from April to July, with peak spawning in June. The study also explored fecundity-related traits, finding that individual absolute fecundity ($F_A$) ranged from 11,364 eggs to 56,377 eggs, while eviscerated body weight relative fecundity ($F_W$) ranged from 209 eggs/g to 823 eggs/g. The exploitation rate ($E$) for *X. argentea* was calculated as 0.574, suggesting that the population of *X. argentea* has been overexploited. By revealing previously unknown data on the key life history traits of *X. argentea*, this study has provided valuable insights that are crucial for the development of conservation strategies and policies.

## INTRODUCTION

The Songhua River is the longest tributary of the Amur River in Northeast China, and abundant fishery resources are available (*Xie, 2007*; *Wang et al., 2018*; *Lu et al., 2023*). It is home to large economic fish species such as *Hypophthalmichthys molitrix, Culter*

*alburnus, Elopichthys bambusa,* and *Coregonus ussuriensis,* as well as small and medium-sized fish species such as *Perca fluviatilis, Pseudobagrus ussuriensis, Hemiculter leucisculus, Pseudorasbora parva,* and *Squalidus argentatus* (*Xie, 2007; Lu et al., 2023*). Tangwang River is a significant tributary of the lower reaches of the Songhua River, which is originates in the Lesser Khingan Mountains, spanning Yichun city and Jiamusi city, with a total length of 509 km, a drainage area of 21,245 km$^2$, a total surface water resource of 5.52 billion m$^3$, and an average estuary flow of 184.6 m$^3$/s (*Wang et al., 2018; Xue et al., 2020; Wang et al., 2021a*). The Tangwang River is a typical mountain-stream river, with most of its main stream crossing between peaks and valleys, lush vegetation on both sides, beautiful environment and clear water quality (*Wang et al., 2015; Wang et al., 2021b*). The perennial water temperature is low, which is more suitable for the survival and reproduction of cold-water fish such as *Oncorhynchus keta, Brachymystax lenok, Esox reicherti,* and *C. ussuriensis* (*Wang et al., 2021a; Li et al., 2021*). According to *Ying*'s (*2015*) research, the Tangwang River is home to a diverse fish community consisting of 42 species belonging to 10 different families, among which small and medium-sized fish are dominant, while large fish are relatively scarce. In recent years, the lower reaches of the Tangwang River have experienced significant challenges due to various factors such as overfishing, environmental degradation, and water conservancy construction (*Liu, 2011; Wang et al., 2018; Wang et al., 2021a*). These issues have had adverse effects on the population of large and formidable cold-water fish in the region. Additionally, migratory species, such as *O. keta,* have encountered such dire circumstances that they are currently on the verge of extinction (*Wang et al., 2021a*).

Fish of the genus *Xenocypris,* belonging to Cypriniformes, are important economic fish, such as *Xenocypris argentea, X. microlepis,* and *X. davidi* (*Xie, 2007; Zhang, Hou & Liu, 2014; Peng et al., 2018; Zhu et al., 2022*). These fish have a unique ecological status and a robust propensity for population increase, which presents enormous opportunities for ecological and commercial benefits from aquaculture (*Xie, 2007; Zhang, Hou & Liu, 2014*). Among which, *X. argentea* is widely distributed in major river systems and their affiliated lakes in China, while abroad it is only found in Russia and Vietnam (*Xiao, Zhang & Liu, 2001; Xie, 2007; Peng et al., 2018; Hu et al., 2019*). It can adapt to complex and diverse habitats, and there are different degrees of geographical isolation, forming different geographical populations (*Liu et al., 2015*). Due to its preference for humus, debris, sapropelic mud, diatoms and epiphytic algae, *X. argentea* is often used as a tool fish in aquaculture and fisheries to help improve water quality and optimize the aquatic ecosystem in reservoirs and ponds (*He et al., 2013*). With environmental pollution, water eutrophication, overfishing and other reasons in recent years, the natural population of *X. argentea* is also declining rapidly (*Zhao, Peng & Guo, 2018*). Currently, research on *X. argentea* primarily centers around aspects such as genetic diversity (*Liu et al., 2015*), geographical distribution (*He et al., 2013*), genetic breeding (*Zhao, Peng & Guo, 2018*), and population differentiation (*Hu et al., 2012*). However, data on its ecology and biology are relatively scarce and limited to southeastern China, even if some studies have been carried out on small-scale genetic ecology (*Hu et al., 2012*).

All aspects of a fish's life history are interconnected. As fish age, their body accumulates metabolites, resulting in continuous changes in length and weight (*Kiani, Keivany & Paykan-Heyrati, 2016*; *Al-Husaini et al., 2021*; *Longo et al., 2021*; *Samejima & Tachihara, 2022*). Adaptively, the growth patterns and behaviors of fish evolve at different stages and are influenced by external environmental factors and genetic traits, enabling them to thrive (*Labbaci, Chaoui & Kara, 2019*; *Kutsyn & Samotoy, 2022*). Through reproduction, the growth process of fish provides energy and material resources, which are then passed on to future generations (*Wieland, 2000*; *Smith & Walker, 2004*; *Thorsen et al., 2010*; *Taylor & Cruz, 2017*). Consequently, the study of age, growth, and reproductive characteristics has long been a focal point in fish population ecology and conservation biology research.

Understanding fundamental biological characteristics and population dynamics relies heavily on important parameters such as age, growth, maturity, fecundity, and mortality (*Frisk, Miller & Fogarty, 2001*; *Pecuchet et al., 2017*; *Al-Husaini et al., 2021*; *Samejima & Tachihara, 2022*). These parameters serve as crucial data for effective fishery management by helping to determine the health of a population, estimate sustainable harvest levels, and develop appropriate conservation measures (*Cao et al., 2009*; *Lu et al., 2023*). In the Tangwang River, a pioneering and in-depth examination of the life-history characteristics of the economically important fish species *X. argentea* was conducted, representing the first comprehensive investigation of this species. This study delved into numerous life-history characteristics, exploring how this species adapts to environmental stress within their habitat. The aims of the article are to offer a comprehensive evaluation of the biological characteristics of *X. argentea*, including age, growth, maturity, fecundity, and mortality. The findings of this study will contribute to an enhanced understanding of the population ecology of *X. argentea*, thus facilitating the assessment and management of crucial fishery resources. Ultimately, this research endeavors to promote sustainable utilization practices.

## MATERIALS AND METHODS

### Sampling

In total, 1,003 specimens of *X. argentea* (474 females and 529 males) were collected between April and November 2022 from the lower reaches of the Tangwang River (129°40′–129°50′E, 46°41′–46°42′N) using gill nets with a mesh size of 3–5 cm, and the samples quickly died naturally after being removed from the fishing nets. Routine biological measurements were carried out on samples in the fresh state, including the total length ($L$, accurate to 0.1 cm) and body weight ($W$, accurate to 0.01 g). Biological anatomy was assessed, and the sex and gonad stage of the specimens were assessed and documented immediately. Additionally, the gonad weight ($GW$) and eviscerated body weight ($EW$) were measured with an accuracy of 0.01 g. Paired sagittal otoliths of each sample were extracted, cleaned with distilled water, and stored dry in labeled tubes. During the survey, water temperature was measured by a portable water quality analyzer (HACH, Loveland, Colorado, USA). The samples were analyzed in accordance with the guidelines of the Heilongjiang River Fisheries Research Institute of CAFS Application for Laboratory Animal Welfare and Ethical review, Harbin, China (Issue No.: 20211210–001).

## Otolith processing and aging

The sagittal otoliths were taken back to the Fisheries Resource Biology Laboratory at the Heilongjiang Fishery Research Institute for further processing. They were fixed, polished, and made transparent through a series of steps. (1) Fixation: The convex side of the otolith was placed facing upward and securely attached to a glass slide using hot melt adhesive. The sample was left to air-dry naturally for 24 h. (2) Polishing: The otolith was polished using progressively finer water mill sandpapers in the following order: 1,500–3,000 grit. Throughout the process, the otolith was observed under a microscope to ensure the desired results. As the polishing approached the core, sandpaper with a higher mesh size was used until the otolith core was clearly visible. (3) Transparency: The otolith was ground down to a thickness of 0.1–0.2 mm, ensuring that the otolith core and peripheral growth rings were clearly visible. It was then made transparent by soaking it in xylene and sealing it with neutral gum. (4) Observation: The otolith was observed under a optical microscope, and photos were taken by the microphotograph system. To ensure accuracy in the analysis, each otolith was read twice by the researchers. If there was any discrepancy between the two counts, the otolith was recounted until a consistent result was obtained (*Wang et al., 2022*).

## Growth characteristics

The power formula was used in the regression analysis to examine the length-weight relationships (LWRs) of *X. argentea* (*Froese, Tsikliras & Stergiou, 2011*). The equation is $W = aL^b$, where $W$ is the body weight (g); $L$ is the total length (cm); $a$ is the conditional factor of growth; and $b$ is the allometric growth index. To examine whether there were statistically significant differences between the sexes, an analysis of covariance was performed using logarithmically converted total length and body weight data. A $t$ test was used to evaluate the differences between the value of $b$ and the value of "3" (indicating an isometric growth pattern).

The von Bertalanffy growth formula (VBGF) was used to fit the growth characteristics of *X. argentea*, which has the following formula: $L_t = L_\infty \left[ 1 - e^{-K(t-t_0)} \right]$, where $L_t$ is the total length at age $t$ (cm); $L_\infty$ is the asymptotic total length (cm); $K$ is growth coefficient ($yr^{-1}$); $t$ is the age of the sample (yr); and $t_0$ is the theoretical initial age at which the total length is zero (yr) (*Von Bertalanffy, 1938*). The growth characteristic index ($\varphi$) was calculated using the following formula: $\varphi = \lg K + 2 \lg L_\infty$, where $K$ and $L_\infty$ are parameters from the VBGF. The growth inflection point age ($t_i$) was calculated using the following formula: $t_i = \ln b / K + t_0$, where $K$ and $t_0$ are parameters from the VBGF, and $b$ is a power exponential coefficient from the LWRs formula. Additionally, the residual sum of squares (ARSS) was used to statistically compare the fitted growth curves between sexes (*Chen, Jackson & Harvey, 1992*). The statistical analyses were carried out using Microsoft Excel 2016 (Microsoft, Redmond, WA, USA) and SPSS Statistics 19.0 (SPSS Inc., Chicago, IL, USA).

## Reproduction

The reproductive period was determined by analyzing the temporal variations in the gonadal somatic index (*GSI*, %). It was calculated using the following formula:

$GSI = GW/EW \times 100$(*Li et al., 2015*). In this formula, *GW* refers to gonad weight, and *EW* refers to eviscerated body weight. Specifically, one-way analysis of variance was used to assess whether there were significant differences in *GSI* values across the months.

The reproductive capacity of the population was assessed using the gravimetric method according to *Li et al. (2015)*, and the maturity stage of female individuals was determined using a visual evaluation according to six scales: stages I, II, III, IV, V, and VI (*Yin, 1993*; *Walsh, Pease & Booth, 2003*). A portion of the eggs (0.1–0.5 g) from the ovaries of female individuals at maturity stages IV–V was collected. These eggs were then stored in a 10% formalin solution. The egg numbers (*N*) of all deposited yolk were counted under a microscope in the laboratory, and the absolute fecundity (*$F_A$*) and relative fecundity of all individuals were calculated by taking the average value. The formulas for evaluating fecundity are as follows:

$$F_A = (N/W_s) \times GW$$
$$F_L = F_A/L$$
$$F_W = F_A/EW$$
$$RCR = (SF/TF) \times 100.$$

In these formulas, *L* is the total length (cm); *EW* is the eviscerated body weight; *GW* is the gonad weight; *N* is the number of sample eggs (eggs); *$W_S$* is the weight of the sample eggs (g); *$F_A$* is the individual absolute fecundity (eggs); *$F_L$* is the relative fecundity of total length; *$F_W$* is the relative fecundity of eviscerated body weight; *SF* is the total fecundity of a single age group; *TF* is the total fecundity of total samples; and *RCR* is the eproductive contribution rate (%).

## Mortality estimation

Age-based catch curve analysis was used to assess the total mortality rate (*Z*) of *X. argentea* (*Ricker, 1975*). The age frequency distribution data are used to fit a linear regression model ($y = mx + n$) to the plotted data points, where the absolute value of the descending slope (*m*) of the regression line represents *Z*. The slope of the line represents the decrease in log-transformed fish numbers as age increases, reflecting the cumulative effects of natural mortality, predation, and fishing.

The natural mortality (*M*) was evaluated by three empirical equations: (1) $\ln M = -0.0066 - 0.279 \ln L_\infty + 0.6543 \ln K + 0.4634 \ln T$ (*Pauly, 1980*), where *K* and *$L_\infty$* are parameters from the von Bertalanffy growth equation, *T* represents the annual habitat temperature (°C) of the water where the fish stocks reside; (2) $M = -0.0021 + 2.5912/t_{max}$ (*Zhan, Lou & Zhong, 1986*), where *$t_{max}$* is the maximum age in the catch; and (3) $M = 0.0189 + 2.06K$ (*Ralston, 1987*), where *K* represents the growth coefficient from the VBGF. The fishing mortality (*F*) was determined using the formula $F = Z - M$, while the exploitation rate (*E*) was estimated as $E = F/Z$.

## RESULTS

### Population structure

Based on the data obtained, the sampled female *X. argentea* exhibited a total length ranging from 12.4 cm to 25.7 cm, with an average length of $18.23 \pm 2.45$ cm. Their body weight ranged from 15.86 g to 159.55 g, with an average weight of $59.25 \pm 25.06$ g. On the other hand, the males had a total length ranging from 10.8 cm to 23.9 cm, with an average length of $17.44 \pm 2.08$ cm. Their body weight ranged from 9.27 g to 121.06 g, with an average weight of $49.67 \pm 17.67$ g. The majority of the females (90.51%) had a total length ranging from 14.0 cm to 22.0 cm, whereas most of the males (85.07%) had a total length ranging from 14.0 cm to 20.0 cm, as depicted in Fig. 1. One-way analysis of variance results showed significant differences between males and females in terms of their total length and body weight. The total length of males was found to be significantly smaller than that of females ($F = 30.545$, $P < 0.01$). Similarly, the body weight of males was also significantly smaller than that of females ($F = 49.730$, $P < 0.01$).

### Age structure

The sagittal otoliths were used for age estimation of *X. argentea*, and the results showed that the ages ranged from 1 to 5 years old in both females and males (Fig. 2, Table 1). In all, 68.78% of the female samples were in the age ranges 2 and 3, while 78.64% of the male samples were in the same range. The number of age 1 individuals was relatively small, which could be attributed to the mesh size of the gill net used. Additionally, the proportion of total samples between the ages of 4 and 5 was quite low, suggesting that the *X. argentea* population's age structure was sampled.

### Growth characteristics

The LWRs of *X. argentea* between sexes showed significant differences based on covariance analysis ($F = 5.146$, $P = 0.024$). The regression functions of LWRs were expressed as: $W_\female = 0.0063 L^{3.1296}$ ($R^2 = 0.9473$, $n = 474$) and $W_\male = 0.0083 L^{3.0274}$ ($R^2 = 0.9375$, $n = 529$) (Fig. 3). The *b* value for females (3.1296) was significantly greater than "3" (*t* test, $t = 5.216$, $P = 1.4 \times 10^{-7}$), indicating a positive allometric growth. For males, there was no significant difference observed between the *b* value (3.0274) and "3" (*t* test, $t = 1.161$, $P = 0.123$), indicating an isometric growth pattern.

The *L*-Age relationship in this study was described using the VBGF. The total length growth formulas were $L_t = 38.161\left[1 - e^{-0.113(t+2.985)}\right]$ ($R^2 = 0.803$) for females and $L_t = 29.248\left[1 - e^{-0.181(t+2.533)}\right]$ ($R^2 = 0.726$) for males. The ARSS test showed no significant difference between sexes ($F = 0.2659$, $P = 1.305$). The total length growth equation for total samples was $L_t = 36.096\left[1 - e^{-0.121(t+3.017)}\right]$ ($R^2 = 0.772$), where the $L_\infty$, $K$ and $t_0$ values were 36.096 cm, 0.121 $yr^{-1}$, and $-3.071$ yr, with standard errors of 4.520, 0.031, and 0.413 and 95% confidence intervals of [27.225–44.966], [0.061–0.181], and [ $-3.828$ to $-0.207$], respectively. The curve of the total length growth fitted by the VBGF is shown in Fig. 4. Additionally, the $\varphi$ value for *X. argentea* was determined to be 2.198, and the $t_i$ value was determined as 6.320 yr.
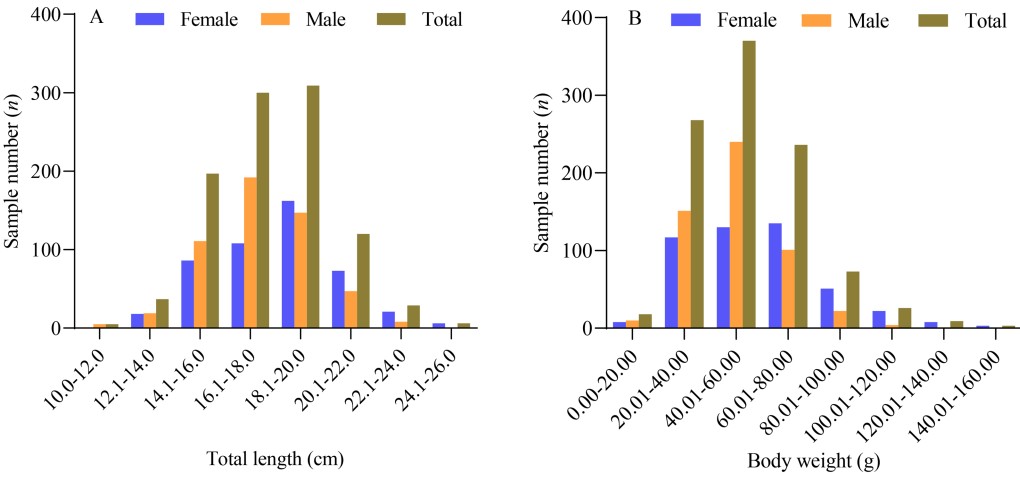

**Figure 1** The distribution of total length (A) and body weight (B) of *X. argentea* in the lower reaches of the Tangwang River.

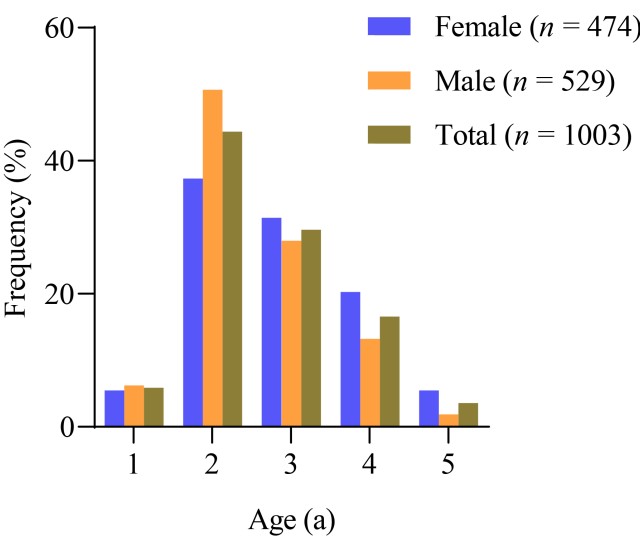

**Figure 2** Frequency distribution of the age of *X. argentea* in the lower reaches of the Tangwang River.

## Maturation

A total of 1,003 fish were studied, and the overall sex ratio (female: male) was 1: 1.17 for *X. argentea*. The *GSI* was calculated monthly for females and males in the study (Fig. 5). The calculated *GSI* values for males ranged from 0.03% to 30.58%, with an average of 1.32 ± 1.92%. In comparison, the *GSI* values for females showed a wider range, varying from 0.10% to 46.51%, with an average of 5.51 ± 7.60%. The highest average values of *GSI* were noted for June with 13.33 ± 9.95% for females and 2.49 ± 1.68% for males, and these values were significantly higher than those in other months ($P$ <0.05). Additionally, the lowest average value of *GSI* was observed in September for females, with a value of 1.17 ± 0.52%.

**Table 1 Numbers of samples and total length ($L$) and body weight ($W$) in different ages of *X. argentea* in the lower reaches of Tangwang River.**

| Sex | Age | $n$ | $L$ (cm) | | $W$ (g) | |
| --- | --- | --- | --- | --- | --- | --- |
| | | | Range | Mean ± SD | Range | Mean ± SD |
| Female | 1 | 26 | 12.4–14.7 | 13.75 ± 0.71 | 15.86–30.67 | 22.91 ± 4.31 |
| | 2 | 177 | 13.8–19.2 | 16.39 ± 1.23 | 22.29–61.34 | 40.08 ± 9.23 |
| | 3 | 149 | 16.4–21.8 | 18.88 ± 0.91 | 38.80–88.22 | 63.10 ± 9.54 |
| | 4 | 96 | 18.87–23.3 | 20.50 ± 1.03 | 59.65–117.68 | 82.08 ± 12.01 |
| | 5 | 26 | 21.5–25.7 | 23.08 ± 1.20 | 99.87–162.31 | 119.93 ± 17.40 |
| Male | 1 | 33 | 10.8–15.7 | 13.57 ± 1.31 | 9.27–32.04 | 21.26 ± 6.36 |
| | 2 | 268 | 13.2–18.8 | 16.47 ± 1.19 | 22.00–63.23 | 40.71 ± 8.54 |
| | 3 | 148 | 15.7–20.5 | 18.40 ± 0.94 | 35.15–75.77 | 56.68 ± 8.18 |
| | 4 | 70 | 18.5–21.9 | 20.21 ± 0.85 | 58.15–93.52 | 74.84 ± 7.92 |
| | 5 | 10 | 21.9–23.9 | 22.44 ± 0.61 | 92.33–121.06 | 103.61 ± 10.66 |
| Total | 1 | 59 | 10.8–15.7 | 13.65 ± 1.08 | 9.27–32.04 | 21.99 ± 5.57 |
| | 2 | 445 | 13.2–19.2 | 16.44 ± 1.21 | 22.00–63.23 | 40.46 ± 8.81 |
| | 3 | 297 | 15.7–21.8 | 18.64 ± 0.95 | 35.15–88.22 | 59.90 ± 9.44 |
| | 4 | 166 | 18.5–23.3 | 20.38 ± 0.97 | 58.15–117.68 | 79.02 ± 11.05 |
| | 5 | 36 | 21.5–25.7 | 22.90 ± 1.01 | 92.33–162.31 | 115.40 ± 17.33 |

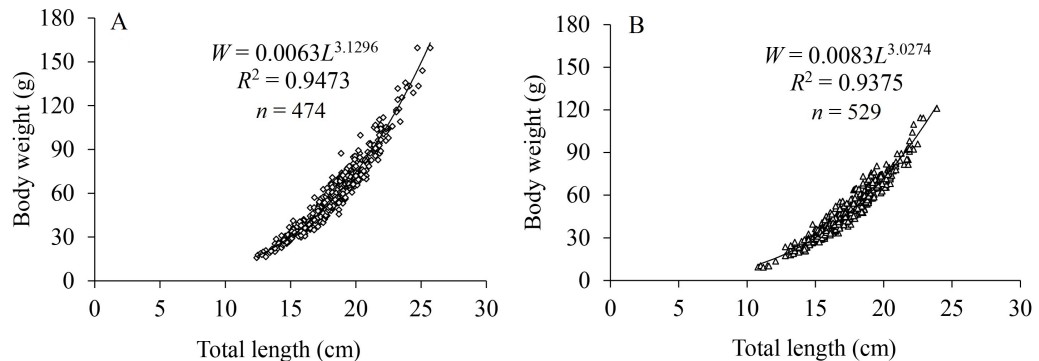

**Figure 3 Length-weight relationships for females (A) and males (B) of *X. argentea* in the lower reaches of the Tangwang River.**

Similarly, for males, the lowest average value of *GSI* was recorded in August, with a value of 1.17 ± 0.52%. According to the results of the study, the gonads of both females and males underwent rapid development from April to June. After June, it was noted that the *GSI* values declined rapidly from June to either August (for females) or September (for males). Subsequently, from either August or September, the *GSI* values gradually increased. Based on the monthly fluctuations in the proportion of ovaries at various developmental stages (Fig. 6), it is evident that the percentage of ovaries in stages IV and V during the months of May, June and July is relatively high, accounting for 36.62%, 68.24%, and 34.18%, respectively. In July, stage VI dominates with a proportion of 21.52%. From August to November, phases II and III emerge as the primary stages. Based on these results, it can be
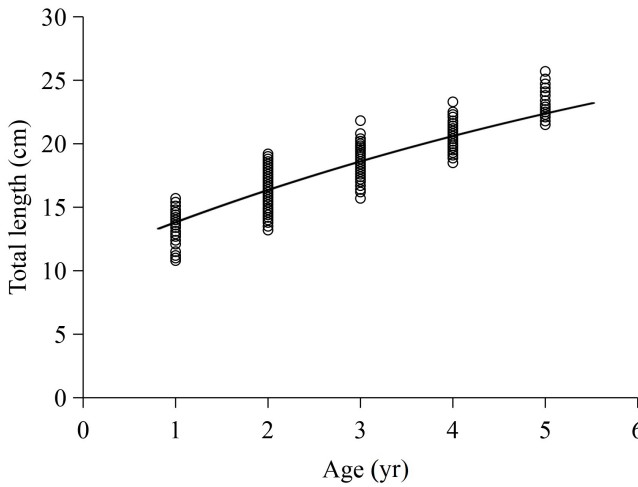

**Figure 4 Von Bertalanffy growth formula fitted to total length-at-age for *X argentea* from samples captured.**

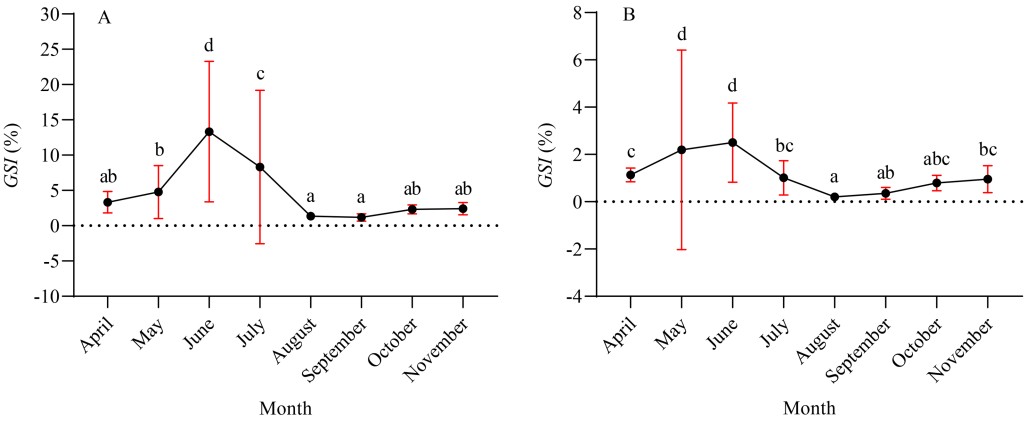

**Figure 5 Mean gonadal somatic index (*GSI*) variation by month for female (A) and male (B) *X. argentea* samples was calculated monthly.** Note: different lowercase letters indicate significant differences (*P* < 0.05).

concluded that the breeding season of *X. argentea* primarily occurs from May to July, with June being the peak season. Furthermore, there was no significant difference (*P* >0.05) in *GSI* between April and November, indicating that gonadal development was almost stagnant during the freezing period (November to April of the next year).

## Fecundity

From June to July, a total of 32 *X. argentea* were randomly selected to evaluate the population fecundity. These samples were divided into four age groups, ranging from 2 to 5 years (Table 2). The contribution rate of each age group to population replenishment can be estimated by comparing the total number of eggs conceived by each age group. The

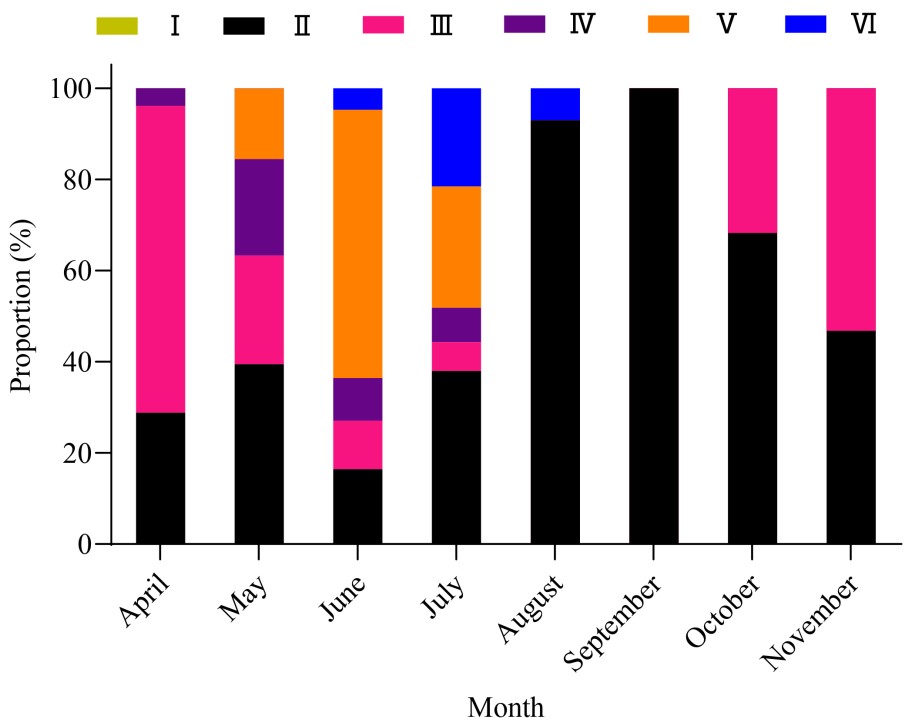

**Figure 6 Variation in maturity stages of ovaries from April to November in female *X. argentea*.**

*RCR* of age 3 is the largest, accounting for 47.53%. It is followed by age 4, accounting for 32.48%. The *RCR* of age 5 is 17.07%, while that of age 2 is the smallest, at only 2.93%. Furthermore, the individual fecundity of *X. argentea*, measured in terms of $F_A$, $F_L$, and $F_W$, shows a gradual increase with age, which suggests that as *X. argentea* individuals grow older, their reproductive capacity in terms of egg production increases (Table 2, Fig. 7A). Additionally, the range of $F_A$ is from 11,364 eggs to 56,377 eggs, of which 65.63% are mainly distributed from 10,000 eggs to 30,000 eggs (Fig. 7B). The range of $F_L$ was from 606 eggs/cm to 2,623 eggs/cm, of which 62.50% were mainly distributed from 600 eggs/cm to 1,400 eggs/cm (Fig. 7C). The range of $F_W$ was from 209 eggs/g to 823 eggs/g, of which 75.00% was mainly distributed from 200 eggs/g to 500 eggs/g (Fig. 7D).

## Mortality and exploitation rate

The survey net mesh proved to be ineffective in capturing the age 1 group, resulting in a limited collection of samples in that age group. As a result, the data from the age 1 group were disregarded for calculating *Z*. Based on the age-based catch curve analysis, the *Z* value for the total samples was determined to be 0.813 yr$^{-1}$, as shown in Fig. 8. The average water temperature (*T*) in the lower reaches of the Tangwang River was 9 °C, so the *M* value for the total samples was estimated as 0.254 yr$^{-1}$ by *Pauly (1980)*, 0.516 yr$^{-1}$ by *Zhan, Lou & Zhong (1986)* and 0.268 yr$^{-1}$ by *Ralston (1987)*. Taking the average of these *M* values, the estimated *M* value for the total samples is determined to be 0.346 yr$^{-1}$. Consequently, the

**Table 2  Characteristics of the biological indices and individual fecundity of *X. argentea* in the lower reaches of the Tangwang River.**

| Item | Age (yr) | | | |
|---|---|---|---|---|
| | 2 | 3 | 4 | 5 |
| $n$ | 2 | 18 | 9 | 3 |
| $L$ (cm) | 17.25 ± 0.58 | 19.29 ± 1.05 | 21.36 ± 0.48 | 24.35 ± 0.71 |
| $W$ (g) | 46.99 ± 6.16 | 72.50 ± 10.04 | 97.46 ± 8.66 | 139.53 ± 17.77 |
| $GW$ (g) | 6.44 ± 0.21 | 12.07 ± 4.72 | 16.33 ± 7.93 | 25.57 ± 8.78 |
| $EW$ (g) | 37.16 ± 7.86 | 54.44 ± 6.88 | 74.05 ± 5.86 | 107.74 ± 8.54 |
| $F_A$ (eggs) | 13,348.15 ± 2,471.84 | 23,843.04 ± 8,043.45 | 32,594.60 ± 13,327.01 | 51,273.98 ± 4,924.41 |
| $F_L$ (eggs/cm) | 771.83 ± 117.35 | 1,239.70 ± 433.61 | 1,522.13 ± 611.45 | 2,103.49 ± 148.75 |
| $F_W$ (eggs/g) | 360.23 ± 9.71 | 442.96 ± 160.76 | 440.89 ± 186.68 | 476.58 ± 41.71 |
| $SF$ (eggs) | 26,696.30 | 429,174.79 | 293,351.42 | 153,821.93 |
| $TF$ (eggs) | | 9,903,044.44 | | |
| $RCR$ (%) | 2.96 | 47.53 | 32.48 | 17.03 |

$F$ value for the total samples was calculated as 0.467 yr$^{-1}$, and the $E$ value was determined to be 0.574.

## DISCUSSION

### Body size of the samples

Residing in the Songhua River basin, *X. argentea* is a stationary species that engages in short-distance river migrations. During spring thaw, they venture into tributaries for breeding and feeding, only to retreat to the river's depths for hibernation before the arrival of autumn and winter freezes. According to *He et al. (2013)*, *X. argentea* can attain a maximum weight of 270 g in natural water bodies, although no specific length data was provided. In this research, the maximum total length and body weight were 25.7 cm and 162.31 g for females, and 23.9 cm and 121.06 g for males. Females in this species exhibited greater body size, with an average total length of 18.23 ± 2.45 cm, compared to males with an average total length of 17.44 ±2.08 cm. This pattern of sexual dimorphism, where females are larger than males, has been observed in other teleosts, such as *Gymnocypris firmispinatus* (*Ma et al., 2019*), *Lateolabrax latus* (*Kunishima et al., 2021*), and *C. ussuriensis* (*Wang et al., 2022*). In this study, the collection of smaller juvenile fish was limited, which may have been caused by the large mesh of fishing nets. A power exponent function was employed to establish the LWRs of *X. argentea*. Studies have shown that several factors can influence the LWRs of fish, such as season, habitat, gonad weight, sex, and stomach fullness (*Froese, Tsikliras & Stergiou, 2011*; *Wang et al., 2022*). In our study, the allometric growth index (*b*) of 3.1296 for females exhibited a significant difference from the value of "3", signifying a positive allometric growth pattern. Conversely, the *b* value for males, which was 3.0274, did not show a significant difference from the value of "3", similar to *X. macrolepis* (3.085) (*Ma, Li & Wu, 2016*), *X. yunnanensis* (2.964) (*Zhang, Wang & Liu, 2017*), *X. fangi* (2.98) (*Ren et al., 2018*), and *X. davidi* (3.07) (*Ren et al., 2018*), which suggests that these *Xenocypris* fishes have an isometric growth pattern. The significant difference in weight

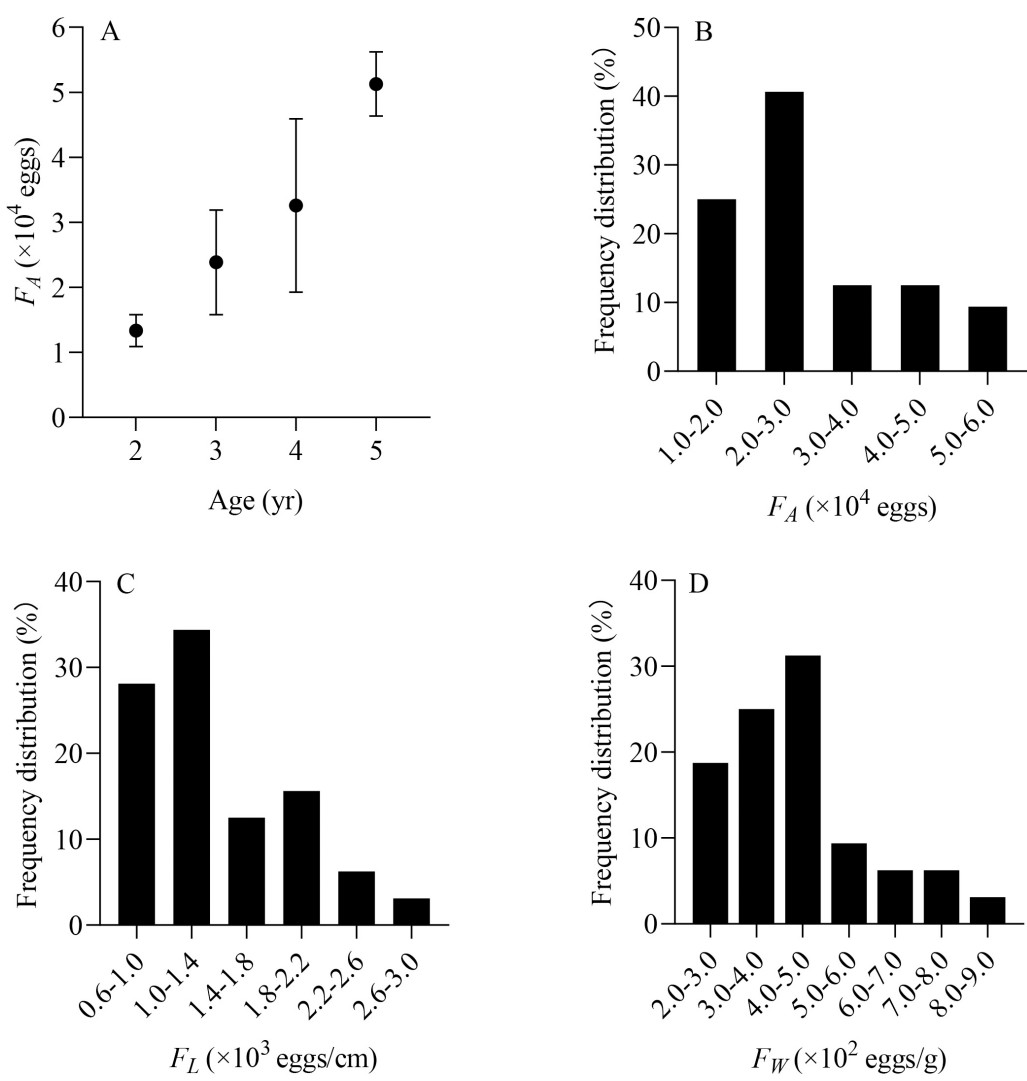

**Figure 7** (A–D) Distribution of individual fecundity for *X. argentea* in the lower reaches of the Tang-wang River. Note: $F_A$ is the individual absolute fecundity (eggs); $F_L$ is the relative fecundity of total length; $F_W$ is the relative fecundity of eviscerated body weight.

between the ovaries in female fish and the testes in male fish during the reproductive period might explain the observed variations in the LWRs between the two sexes of *X. argentea*.

## Age and growth

Accurately acquiring age data is crucial for studying the life history and population ecology of fish, as well as for analyzing and evaluating fish resources (*Al-Husaini et al., 2021*; *Samejima & Tachihara, 2022*). Previous studies have shown that otoliths are an ideal material to record the life history of fish and changes in the surrounding environment because they are not easily reabsorbed and grow continuously (*Ma et al., 2019*; *Al-Husaini et al., 2021*; *Wang et al., 2022*). In this research, we used the otolith to determine the age of *X. argentea*. Our findings revealed that both female and male *X. argentea* had an age

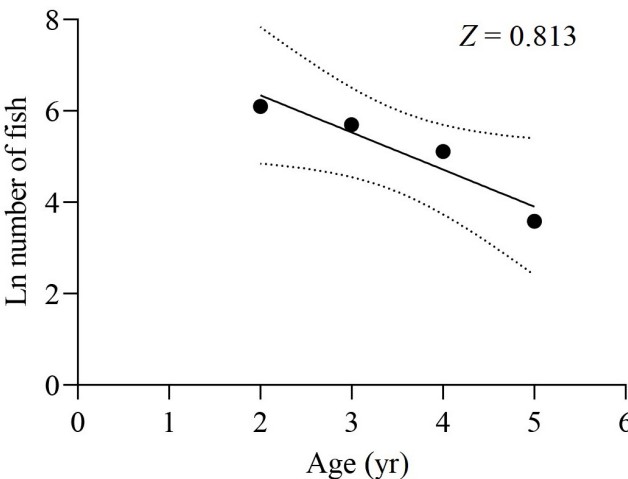

**Figure 8** Catch curve based on observed age for *X. argentea* in the lower reaches of the Tangwang River.

range of 1 to 5 years old. This age distribution is consistent with that of *X. macrolepis* and *X. davidi* in Qiandao Lake (*Zhang, Hou & Liu, 2014*). Growth parameters play a fundamental role in investigating the biological and ecological characteristics of fish (*Boisclair & Leggett, 1989*). Alterations in these parameters can significantly affect the mortality, biomass, and distribution of fish populations (*Rochet & Trenkel, 2003*; *Coulson et al., 2014*). Consequently, they serve as the foundation for constructing fish stock assessment models and evaluating the influence of human activities and environmental factors on fish populations. Growth coefficient $K$ provide insight into the growth rate of fish, which ultimately determines when the fish will reach its asymptotic length $L_\infty$ (*Frisk, Miller & Fogarty, 2001*). The growth coefficient $K$ and growth characteristic index ($\varphi$) are the key parameters used to evaluate the growth potential of fish populations (*Onikura & Nakajima, 2013*). In this study, the growth coefficient $K$ of *X. argentea* was 0.121 yr$^{-1}$. Compared with other fishes in the Songhua River basin (Table 3), the $K$ value of *X. argentea* is higher than that of *P. fluviatilis* (0.074 yr$^{-1}$) (*Xu et al., 2021*) but lower than that of *C. ussuriensis* (0.139 yr$^{-1}$) (*Wang et al., 2022*), *H. molitrix* (0.258 yr$^{-1}$) (*Wang et al., 2020*), *Leuciscus waleckii* (0.260 yr$^{-1}$) (*Lu et al., 2019*), *H. leucisculus* (0.600 yr$^{-1}$) (*Lu et al., 2023*), *P. parva* (0.710 yr$^{-1}$) (*Lu et al., 2023*), and *S. argentatus* (0.710 yr$^{-1}$) (*Lu et al., 2023*). In addition to habitat environmental factors, variations in the $K$ value among fishes may also be attributed to the size of the collected individuals, as the maximum and minimum total lengths within the samples significantly affect the estimation of the growth coefficient $K$ (*Lu et al., 2019*; *Wang et al., 2020*; *Lu et al., 2023*). Based on the value of $\varphi$, it was evident that *C. ussuriensis* (*Wang et al., 2022*), *P. fluviatilis* (*Xu et al., 2021*), *H. molitrix* (*Wang et al., 2020*), *L. waleckii* (*Lu et al., 2019*), and *H. leucisculus* (*Lu et al., 2023*) exhibited faster growth than *X. argentea*. Conversely, *P. parva* and *S. argentatus* (*Lu et al., 2023*) demonstrated slower growth rates than *X. argentea*. These findings reveal that the growth rate of *X. argentea* in the Songhua River basin can be considered relatively slow among these fish species.
**Table 3  Growth parameters of several fishes in the Songhua River basin.**

| Species | $L_\infty$ (mm) | $K$ (yr$^{-1}$) | $t_0$ (yr) | $\varphi$ | Sources |
|---|---|---|---|---|---|
| *Coregonus ussuriensis* | 732.82 | 0.139 | −0.153 | 4.873 | *Wang et al. (2022)* |
| *Perca fluviatilis* | 714.30 | 0.074 | −1.622 | 4.577 | *Xu et al. (2021)* |
| *Hypophthalmichthys molitrix* | 745.55 | 0.258 | 0.113 | 5.157 | *Wang et al. (2020)* |
| *Leuciscus waleckii* | 351.75 | 0.260 | −0.545 | 4.507 | *Lu et al. (2019)* |
| *Hemiculter leucisculus* | 198.20 | 0.600 | −0.270 | 4.370 | |
| *Pseudorasbora parva* | 99.75 | 0.710 | −0.270 | 3.850 | *Lu et al. (2023)* |
| *Squalidus argentatus* | 89.25 | 0.710 | −0.270 | 3.850 | |
| *X enocypris argentea* | 360.96 | 0.121 | −3.017 | 4.198 | This research |

## Reproductive characteristics

Reproduction encompasses a vital aspect of the life history of fish, encompassing various processes such as the sexual system, gonad development, migration, spawning, fertilization, hatching, and the subsequent growth and development of larvae and juveniles (*Grandcourt et al., 2009*). To cope with environmental changes, fishes have developed distinct reproductive strategies, primarily aimed at maximizing the production of offspring within the constraints of limited energy and lifespan (*Kurita, Meier & Kjesbu, 2003*; *McBride et al., 2015*). Based on the average month *GSI*, it was observed that *X. argentea* exhibited a relatively brief spawning period between May and July. The peak of its spawning activity occurred in June, aligning closely with the spawning periods of *X. macrolepis* and *X. davidi* from Qiandao Lake in China (*Zhang, Hou & Liu, 2014*). *Hashiguti, Rocha & Montag (2017)* indicated that water temperature and rainfall are important factors influencing fish spawning. In the Tangwang River, the freezing period lasts from November to April of the subsequent year (*Wang et al., 2015*). After the river thaws, there is a significant and rapid increase in river water temperature during the period from April to June, escalating from 2.0 °C to 18.5 °C. The swift elevation of water temperature in the Tangwang River has encouraged gonad development in various fish species. During this stage, various fish species such as *Gobio cynocephalus*, *Phoxinus lagowskii*, *Perccottus glenii*, *P. parva*, and *Opsariichthys bidens*, among others, actively participate in spawning and breeding activities (*Ying, 2015*). Furthermore, starting from April to July, the rising water temperature triggers an increase in the abundance of bait within the water, such as the abundance of benthic diatoms (*Xue et al., 2020*). Opting to breed *X. argentea* during this period, specifically from May to July, ensures that the offspring have access to ample food sources and an optimal water temperature for their growth, ultimately enhancing their survival rate (*Adams et al., 2018*; *Xue et al., 2020*; *Liu et al., 2022*). *X. argenteus* species typically overwinter in the deep water areas of the Songhua River, from mid-late November to early April of the following year. The similarity in *GSI* values observed between April and November implies that the reproductive development of *X. argenteus* remains stagnant during the freezing period. This phenomenon may be influenced by factors including low temperatures, limited availability of food organisms, and the swift flow rate in the main stream of the Songhua River.

Fecundity plays a crucial role in the reproductive strategies of fish and varies significantly between species and within species (*Murua & Saborido-Rey, 2003*). These differences and changes are an outcome of long-term evolutionary processes, enabling fish to adapt to external environmental conditions (*Thorsen et al., 2010*; *Taylor & Cruz, 2017*). In our study, the average $F_A$ of *X. argentea* was 28,220.14 ± 12,863.06 eggs, which was significantly lower than that of *X. macrolepis* (71,283 ± 25,360 eggs) and *X. davidi* (10,9141 ± 41,070 eggs) in Qiandao Lake (*Zhang, Hou & Liu, 2014*). The average $F_W$ of *X. argentea* was 440.36 ± 154.36 eggs/g, similar to that of *X. davidi* (408 ± 154 eggs/g) and higher than that of *X. macrolepis* (289 ± 64 eggs) (*Zhang, Hou & Liu, 2014*). These results indicated that the population fecundity belonging to various species within the same genus exhibits notable variations, which can be attributed to factors such as sexual maturity age, individual size, egg diameter, habitat environment, and nutritional condition (*Kurita, Meier & Kjesbu, 2003*; *McBride et al., 2015*). Generally, the relative fecundity of weight is indicative of the reproductive strategy employed by fish (*Liu et al., 2018*). *Melo et al. (2011)* indicated that fish species with smaller eggs tend to produce small larvae with lower survival rates, and these species typically maintain population stability by laying a larger number of eggs. On the other hand, fish species with larger eggs produce larger larvae that exhibit higher survival rates but have relatively lower reproductive capacity (*Melo et al., 2011*). Taken together, the reproductive strategy of *X. argentea* typically involves producing a large number of offspring through breeding, aiming to ensure the stability of its population.

## The mortality and exploitation rate of *X. argentea*

According to *Pauly (1980)*, the primary factor influencing fish resource dynamics is fish mortality. *M* is caused by all possible causes of death except fishing, such as fish age, growth conditions and habitat environment (*Voulgaridou & Stergiou, 2003*). In this study, we used three methods to evaluate *M*, including the length-based empirical relationship proposed by *Pauly (1980)*, the age-based method proposed by *Zhan, Lou & Zhong (1986)* and *Ralston (1987)*. The evaluation results obtained through multiple methods ensure the accuracy of the assessment and determine the *M* value of *X. argentea* in the lower reaches of the Tangwang River was 0.346 yr$^{-1}$. *Li & Chen (2009)* suggested that an exploitation rate of 0.5 is considered most suitable for fish populations, with fishing mortality being equivalent to natural mortality (*Sun et al., 2021*). In this study, the *E* value of *X. argentea* was 0.574, which was higher than 0.5, similar to *L. waleckii* in the lower reaches of the Tangwang River (*Lu et al., 2019*) and *S. argentatus* in the lower reaches of the Songhua River (*Lu et al., 2023*), indicating that *X. argentea* has been overexploitated under the current fishing intensity. Nowadays, with the increase in current commercial fishing intensity and the reduction in mesh size, small and medium-sized fish have gradually become the target of fishing and made into fishmeal for profit (*Lu et al., 2023*). As we all know, small and medium-sized fish have a critical role in the primary production of the ecosystem, since they are the main food source for large commercial fish. These fish population fluctuations have the potential to affect not just the commercial fish community structure but also fish populations across the ecosystem.

To ensure the protection of *X. argentea*, it is crucial to establish effective management strategies. The management department should strengthen the protection of fishery resources in the Songhua River basin and actively control the use of small mesh (less than 3 cm) or illegal fishing, and the occurrence of electric fishing and poisoning of fish should be strictly prohibited. Furthermore, the aquatic germplasm resource conservation zone in the lower reaches of the Tangwang River should be established, and fishery fishing should be prohibited during the breeding season. By implementing a series of effective measures, the fishery resources in the Songhua River basin can be effectively restored.

## CONCLUSION

Through our research, we have successfully addressed the information gap pertaining to the essential life history traits of *X. argentea* in the studied area. According to our analysis, both female and male *X. argentea* have a maximum estimated age of 5 years. Compared to other fish species in the Songhua River basin, *X. argentea* exhibited a moderate growth rate. The spawning period of *X. argentea* ranged from April to July, with the peak season occurring in June. Additionally, the reproductive strategy of *X. argentea* involves producing a significant number of offspring through breeding, with the aim of ensuring population stability. Our findings also indicate that the population of *X. argentea* has been overexploited under the current fishing intensity, so the fisheries management department should strengthen law enforcement efforts to regulate and control illegal fishing activities during the fishing ban period.

### Funding

The research was financially supported by the Special project on agricultural financial fund from the Ministry of Agriculture and Rural Affairs of China entiled ''Survey of fishery resources and environment in key waters of Northeast China'' and Central Public-interest Scientific Institution Basal Research Fund (No. 2020TD07). The funders had no role in study design, data collection and analysis, decision to publish, or preparation of the manuscript.

### Grant Disclosures

The following grant information was disclosed by the authors:
Ministry of Agriculture and Rural Affairs of China.
Central Public-interest Scientific Institution Basal Research Fund: 2020TD07.

### Competing Interests

The authors declare there are no competing interests.

### Author Contributions

- Peilun Li conceived and designed the experiments, performed the experiments, analyzed the data, prepared figures and/or tables, authored or reviewed drafts of the article, and approved the final draft.

- Jiacheng Liu performed the experiments, analyzed the data, authored or reviewed drafts of the article, and approved the final draft.
- Wanqiao Lu performed the experiments, authored or reviewed drafts of the article, and approved the final draft.
- Shuyang Sun performed the experiments, authored or reviewed drafts of the article, and approved the final draft.
- Jilong Wang conceived and designed the experiments, performed the experiments, analyzed the data, prepared figures and/or tables, authored or reviewed drafts of the article, and approved the final draft.

## Animal Ethics

The following information was supplied relating to ethical approvals (i.e., approving body and any reference numbers):

All samples were conducted in accordance with the guidelines of Heilongjiang River Fisheries Research Institute of CAFS Application for Laboratory Animal Welfare and Ethical review (Issue No.: 20211210-001).

## Data Availability

The raw data is available in the Supplementary File.

## Supplemental Information

Supplemental information for this article can be found online at http://dx.doi.org/10.7717/peerj.16673#supplemental-information.

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
