# Peer review of "Age, growth, reproduction and mortality of Xenocypris argentea (Günther,1868) in the lower reaches of the Tangwang River, China"

_PeerJ, doi:10.7717/peerj.16673_

## Round 0.1 · original submission · Minor Revisions

Dear Authors

The reviewers have commented on your manuscript. You can find the attached reports. Based on the comments and suggestions of the expert reviewers, a minor revision is needed for your article.

I would like to request that you check and correct the manuscript step by step based on the reports.

Best regards

Reviewer 1 ·

Basic reporting

My general comments as follow:
1. The paper is very well written; however, it is not easy to follow. This version is long. I suggest if possible, shortening it by authors.
2. There are a lot of spelling mistakes in the figures, all of them should be corrected (and in the text as well).
3. Some references used in the discussion, should be used in the introduction as well.
4. More information should be given at the introduction part related to other similar fish species.
5. If there is a previous study about the species on the same water source, it should be mentioned.
6. Please note that I didn`t check the references.

Experimental design

Please see attached.

Validity of the findings

Please see attached.

Additional comments

Please see attached.

Annotated reviews are not available for download in order to protect the identity of reviewers who chose to remain anonymous.

Reviewer 2 ·

Basic reporting

I think this research is well written and meets the journal's subject.

Line 32. Please use keywords which are not already given in the tittle
Line 127. “an” instead of “a”
Line 162-163. “The 163 statistical analyses were carried out using the Microsoft Excel 2016 and SPSS Statistics 19.0.” remove this, already given above.
Line 221. “an” instead of “a”
Line 360. “while” start with upper case
Line 386. “Nowadays” instead of “Nowdays”
Line 393. “Furthermore” instead of “Foethermore”
Figure 4. Please provide the number of the indivuduals were analysed.

Experimental design

I think the MS meets the journal standards, with suggested improvements.
Only issue I would like to emphasis about the name of the species.
In Eschmeyer's Catalog of Fishes the species listed as synonym of X. macrolepis. I am aware the species has a complex nomenclature story.

There are 2 argenteus in the genus Xenocypris.
1) Leuciscus argenteus Basilewski 1855, but this one is not available because it is a primary junior homonym of Leuciscus argenteus Fitzinger, 1832 and Leuciscus argenteus Storer, 1839. It cannot be used.
2) Xenocypris argentea Günther, 1868 is also not available. It is a secondary junior homonym of L. argenteus Basilewski 1855 when L. a. Basilewski was placed in Xenocypris by Sauvage & Dabry de Thiersant, 1874. It cannot be used.
The next name proposed for this species is X. macrolepis, Bleeker 1851 and this is the valid name as mentioned by authors. However, I think the authors should mention the situation above and clarify that the species remains valid as X. macrolepis in the M & M section.
To see the following publication will be helpful:
Kottelat, M. (2013). The fishes of the inland waters of Southeast Asia: a catalogue and core bibliography of the fishes known to occur in freshwaters, mangroves and estuaries. Raffles Bulletin of Zoology.
"page 170"
Kottelat, M. (2001). Freshwater fishes of northern Vietnam. World Bank, Washington.
"page 44"

Validity of the findings

I believe that the findings of the study are realistic and sufficient.

Additional comments

I accepted the MS because I believe the authors will take into consideration my very few suggestions.

---

## Round 0.2 · accepted · Accept

Dear Dr. Li

I would like to thank you and your co-authors for making the corrections and changes requested by the reviewers. I read and checked carefully your valuable article and I am happy to inform you that your article has been accepted for publication in PeerJ.

Best regards